# Serological Cross-Reactions between Expressed VP2 Proteins from Different Bluetongue Virus Serotypes

**DOI:** 10.3390/v13081455

**Published:** 2021-07-26

**Authors:** Petra C. Fay, Fauziah Mohd Jaafar, Carrie Batten, Houssam Attoui, Keith Saunders, George P. Lomonossoff, Elizabeth Reid, Daniel Horton, Sushila Maan, David Haig, Janet M. Daly, Peter P. C. Mertens

**Affiliations:** 1School of Veterinary Medicine and Science, University of Nottingham, Sutton Bonington, Loughborough LE12 5RD, UK; Petra.Fay@Pirbright.ac.uk (P.C.F.); Elizabeth.Reid@Nottingham.ac.uk (E.R.); David.Haig@Nottingham.ac.uk (D.H.); janet.daly@Nottingham.ac.uk (J.M.D.); 2The Pirbright Institute, Surrey, Woking GU24 ONF, UK; carrie.batten@pirbright.ac.uk; 3UMR VIROLOGIE 1161, INRAE, Ecole Nationale Vétérinaire d’Alfort, ANSES, Université Paris-Est, F-94700 Maisons-Alfort, France; fauziah.mohd-jaafar@vet-alfort.fr (F.M.J.); houssam.attoui@vet-alfort.fr (H.A.); 4John Innes Centre, Department of Biochemistry and Metabolism, Norwich Research Park, Norwich NR4 7UH, UK; keith.saunders@jic.ac.uk (K.S.); george.lomonossoff@jic.ac.uk (G.P.L.); 5Pathology and Infectious Diseases, School of Veterinary Medicine, University of Surrey, Guildford GU2 7XH, UK; d.horton@surrey.ac.uk; 6Department of Animal Biotechnology, Lala Lajpat Rai University of Veterinary & Animal Sciences, Hisar 125004, India; sushilamaan105@googlemail.com

**Keywords:** Bluetongue virus, BTV, orbivirus, orbivirus serotypes, cross-serotype antibodies, VP2, plant expressed proteins, antigenic cartography

## Abstract

Bluetongue (BT) is a severe and economically important disease of ruminants that is widely distributed around the world, caused by the bluetongue virus (BTV). More than 28 different BTV serotypes have been identified in serum neutralisation tests (SNT), which, along with geographic variants (topotypes) within each serotype, reflect differences in BTV outer-capsid protein VP2. VP2 is the primary target for neutralising antibodies, although the basis for cross-reactions and serological variations between and within BTV serotypes is poorly understood. Recombinant BTV VP2 proteins (rVP2) were expressed in *Nicotiana benthamiana*, based on sequence data for isolates of thirteen BTV serotypes (primarily from Europe), including three ‘novel’ serotypes (BTV-25, -26 and -27) and alternative topotypes of four serotypes. Cross-reactions within and between these viruses were explored using rabbit anti-rVP2 sera and post BTV-infection sheep reference-antisera, in I-ELISA (with rVP2 target antigens) and SNT (with reference strains of BTV-1 to -24, -26 and -27). Strong reactions were generally detected with homologous rVP2 proteins or virus strains/serotypes. The sheep antisera were largely serotype-specific in SNT, but more cross-reactive by ELISA. Rabbit antisera were more cross-reactive in SNT, and showed widespread, high titre cross-reactions against homologous and heterologous rVP2 proteins in ELISA. Results were analysed and visualised by antigenic cartography, showing closer relationships in some, but not all cases, between VP2 topotypes within the same serotype, and between serotypes belonging to the same ‘VP2 nucleotype’.

## 1. Introduction

*Bluetongue virus* is the ‘type’ species and most intensively studied member of the genus *Orbivirus*, within the family *Reoviridae*, order *Reovirales* [1,2,3,4]. BTV can infect most ruminants, as well as camelids and some large carnivores [5], causing ‘bluetongue’ (BT), an economically important, clinically severe and sometimes fatal disease primarily of sheep, cattle and some deer species [5,6]. At least 28 BTV serotypes have been identified in serum neutralisation tests (SNT), most of which are transmitted by adult females of vector-competent biting-midge species (*Culicoides* spp.) [6,7,8]. However, some of the most recently discovered BTV serotypes [9,10,11,12,13] do not appear to infect midges and are thought to be transmitted by direct contact between individual hosts [7,14]. 

In recent decades, bluetongue has spread into new habitats, including 12 serotypes detected in Europe since 1998. This movement has been linked to increased human travel and trade, and the effects of climate change on vector-insect activity and distribution in the region [15,16,17]. 

BTV is usually regarded as having a non-enveloped, icosahedral virus particle, although membrane-enveloped virus particles (MEVP) have also been observed by electron-microscopy [1,2,3,4]. The BTV capsid is approximately 80 nm in diameter, comprising three concentric layers of proteins. The innermost ‘sub-core’ layer, which is composed of virus-protein 3 (VP3), encoded by the third largest of the ten BTV genome segments (Seg-3), is surrounded by an intermediate ‘core-surface’ layer of VP7 (encoded by Seg-7), with an ‘outer-capsid’ composed of VP2 and VP5 (encoded by Seg-2 and Seg-6 respectively) [1,2,3]. The BTV core contains ten linear double-stranded (ds) RNA genome segments, associated with multiple transcriptase complexes, composed of minor proteins VP1, VP4 and VP6 (encoded by Seg-1, -4 and -9) [1,2]. BTV also encodes at least five distinct non-structural proteins [4,18]. The outer-most BTV capsid protein, VP2, displays hemagglutination activity and is responsible for cell attachment during the early stages of infection [19,20]. 

Interaction of VP2 with the antibodies generated by infected mammalian hosts can block cell binding, neutralising BTV infectivity [21,22,23,24]. The specificity of these neutralising antibodies (nAbs) (as detected in SNT) is controlled by variations in the amino acid (aa) sequence of VP2 [24,25,26,27]. Phylogenetic analyses of Seg 2 (which encodes VP2) shows a strong correlation with the serological identity of BTV isolates, with between 29 and 59% nucleotide sequence variations between different serotypes [24,25,26,27]. These phylogenetic analyses have also identified closer relationships between Seg-2 of some BTV serotypes, placing them in 11 larger groups (‘Seg-2 nucleotypes’ A–K) [20,24,27]. The nAbs targeting BTV VP2 are protective, and consequently, VP2 is a primary target for vaccine development. Seg-2 has also become a target for the RT-PCR assays that are now widely used for rapid diagnosis, detection and identification of different BTV serotypes in epidemiological and ‘vaccine matching’ studies, largely replacing the slower and less-sensitive SNT [20,28,29,30,31]. 

However, up to 32% sequence variations can exist in Seg-2, with variations in VP2 aa sequence up to 16%, between viruses within the same BTV serotype. These intra-serotype differences often reflect different geographical origins, grouping isolates of the same serotype from south-east Asia, India, China and Australia into a major eastern Seg-2/VP2 ‘topotype’, while viruses from Africa, and North and South America form a major western Seg-2/VP2 ‘topotype’. However, recent intercontinental movements and spread of BTV strains are increasingly blurring these geographic separations [20,32,33]. 

In addition to the strong serotype-specific reactions, low-level, variable, or one-way cross-serotype reactions have also been detected in cross-protection studies in sheep and in tissue culture based SNT, that show at least partial correlation with the ‘nucleotype grouping’ of BTV Seg-2 [25,27,34,35]. However, nAbs are only a subset of the antibodies generated against VP2 and the other structural and non-structural proteins of the virus during infection of mammalian hosts. A significant proportion of the VP2-binding Abs (VP2-bAbs) are non-neutralising and may be both non-protective and more cross-reactive between serotypes [36,37]. 

In order to explore the potential for serotype cross-reactive antibodies and vaccines targeting VP2, we have expressed rVP2s from 17 BTV strains in *Nicotiana benthamiana* [23,38,39]. These include proteins from isolates of 11 serotypes detected in Europe and the Mediterranean region since 1998 [15,40,41,42], as well as from different topotypes of four serotypes and isolates of novel serotypes BTV-25, -26 and -27) (Table 1). Ten of these rVP2 proteins were used to generate antisera in rabbits and the specificity of VP2-bAbs was analysed by indirect (I)-ELISA. The results were visualised by antigenic-cartography and compared to data generated in using reference antisera from sheep previously infected with strains of the different BTV serotypes. The subset of VP2 specific neutralising antibodies (nAbs) was also evaluated in SNT against the reference strains of BTV serotypes-1 to -24, -26 and -27. 

## 2. Materials and Methods

### 2.1. BTV Seg-2 and VP2 Sequence Data and Protein Synthesis 

#### 2.1.1. rVP2 Protein Production

The nucleotide sequences of Seg-2 from different BTV strains (Table 1) downloaded from the Genbank database (https://www.ncbi.nlm.nih.gov/genbank/ (accessed on 21 March 2014)), were codon optimised for plant expression, then synthesised by GeneArt (ThermoFisher Scientific, Waltham, MA, USA) with a sequence encoding a 6xHis-tag inserted at the C-terminus (to enable purification by immobilised metal affinity chromatography (IMAC) Qiagen; Hilden Germany), and flanking *Age*I and *Xho*I, UK restriction sites (New England Biolabs, MA, USA) [40]. These DNA constructs were individually cloned into pEAQ-*HT* expression vectors (Kindly provided by G. Lomonossoff (JIC; UK) to generate pEAQ-*HT*-BTV-VP2 plasmids for each BTV strain [43,44]. Plant expression and purification of VP2 proteins have previously been described [23].

#### 2.1.2. Phylogenetic Comparisons of VP2 Proteins and Subdomains

Full-length aa sequences for VP2, or VP2 sub-domains were aligned using the Clustal X programme V 2.1. Alignment files were converted into the MEGA format using MEGA X MEGA V 7.0.26 software. Neighbour-joining phylogenetic trees were constructed in MEGA X, using the p-distance algorithm (pairwise deletion). Bootstrap (500 replications) analysis was used to test the robustness of phylogenetic groupings.

### 2.2. Virus Culture and Titration

Isolates of field and ‘reference’ strains of different BTV serotypes (Table 1) obtained from the Orbivirus Reference Collection (ORC) at the Pirbright Institute (TPI), (https://www.reoviridae.org/dsRNA_virus_proteins/ReoID/BTV-Nos.htm (accessed on 21 March 2014)) were used to infect 80–90% confluent monolayers of baby hamster kidney (BHK) cells, in T75 cm^2^ tissue culture flasks (ThermoFisher Scientific; Waltham, MA, USA). The inoculum was prepared using 4.5 mL of Eagles medium (ThermoFisher Scientific; Waltham, MA, USA), supplemented with 100 IU/mL penicillin, 100 μg/mL streptomycin (ThermoFisher Scientific; Waltham, MA, USA), containing 500 μL of the virus isolate, mixed and added to the cell layer, then incubated at room temperature for 30 min. Cell media (22 mL) was added and flasks were incubated at 37 °C in 5% CO_2_, then monitored daily for cytopathic effect (CPE). From day 5 onwards, cells showing 90% CPE were harvested and centrifuged to pellet cell debris for 5 min at 800× *g*.

Viruses in tissue culture supernatants were titrated in 96-well tissue culture plates (NUNC (ThermoFisher Scientific; Waltham, MA, USA) as 6-well repeats, containing 100 μL of log10 serial dilutions (from 10^−1^ to 10^−7^, including 2 half log10 dilutions at 10^−3.^^5^ and 10^−4.^^5^) in DMEM (ThermoFisher Scientific; Waltham, MA, USA), containing penicillin and streptomycin (100 IU/mL and 100 μg/mL, respectively). The culture medium was used in an uninfected control. Vero cells (50 μL, containing 2 × 10^5^ cells/mL) were added to all wells. Plates were incubated at 37 °C in 5% CO_2_ and analysed for CPE (which are clearly visible as rounded up and detached cells) by inverted light microscopy on day 6 and 7. Readings on day 7 were used for the final calculation of virus titre using the Spearman-Karber formula [45]. 

**Table 1 viruses-13-01455-t001:** BTV strains and VP2 sequences used for VP2 expression, SNT and phylogenetic comparisons.

BTV Serotype and Country/Region, or Reference-Strain Serotype * [ORC Collection No **]	Serotype, Topotype (Nucleotype ^)	Plant Expression & ELISA Antigen	Used to Immunise Rabbits	Virus Used in SNT & Generate Anti-BTV Sheep Reference-Sera	Seg-2 Acc. No. (Gen Bank)	Reference
BTV-1w Gibraltar [GIB2007/06]	1w (H)	Yes	Yes	-	KP821004	[46]
BTV-1 * [RSArrrr/01]	1w (H)	-	-	Yes	AJ585122	[27]
BTV-1e Greece [GRE2001/09]	1e (H)	-	-	Yes	-	-
BTV-1e Greece [GRE2001/06]	1e (H)	Yes	Yes	-	KP821006	[46]
BTV-2 Tunisia [TUN2000/01]	2w (I)	Yes	No	-	KP821037	[45]
BTV-2 * [RSArrrr/02]	2w (I)	-	-	Yes	AJ585123	[27]
BTV-3 * [RSArrrr/03]	3w (B)	-	-	Yes	AJ585124	[27]
BTV-4e China (1996) YTS-4	4e (A)	Yes	No	-	JX560414	[47]
BTV-4w Cyprus [RSArrrr/04]	4w (A)	-	-	Yes	AJ585125	[27]
BTV-4w Morocco [MOR2009/09]	4w (A)	Yes	Yes	-	KP821064	[46]
BTV-5 * [RSArrrr/05]	5w (E)	-	-	Yes	AJ585126	[27]
BTV-6 * [RSArrrr/06]	6w (C)	-	-	Yes	AJ585127	[27]
BTV-6w Netherlands [NET2008/05]	6w (C)	Yes	Yes	-	GQ506473	[40]
BTV-7 * [RSArrrr/07]	7w (F)	-	-	Yes	AJ585128	[27]
BTV-8 * [RSArrrr/08]	8w (D)	-	-	Yes	AJ585129	[27]
BTV-8w Netherlands [NET2008/03]	8w (D)	Yes	Yes	-	KP821074	[45]
BTV-9 * [RSArrrr/09]	9w (E)	-	-	Yes	AJ585130	[27]
BTV-9e India (2002) MBN	9e (E)	Yes	No	-	JF443156	[48]
BTV-9w Libya [LIB2008/03]	9w (E)	Yes	No	-	KP821087	[46]
BTV-10 Portugal [RSArrrr/10]	10w (A)	Yes	No	Yes	AJ585131	[27]
Germany (2010) BTV-11_DE	11w (A)	Yes	Yes	-	JQ972852	[49]
BTV-11 * [RSArrrr/11]	11w (A)	-	-	Yes	AJ585132	[27]
BTV-12 * [RSArrrr/12]	12w (G)	-	-	Yes	AJ585133	[27]
BTV-13 * [RSArrrr/13]	13w (B)	-	-	Yes	AJ585134	[27]
BTV-14 * [RSArrrr/14]	14w (C)	-	-	Yes	AJ585135	[27]
BTV-14 Russia [RUS2011/01]	14w (C)	Yes	Yes	-	KP821096	[46]
BTV-15 * [RSArrrr/15]	15w (J)	-	-	Yes	AJ585136	[27]
BTV-16 * [RSArrrr/16]	16e (B)	-	-	Yes	AJ585137	[27]
BTV-16w Nigeria [NIG1982/10]	16w (B)	Yes	No	-	AJ585150	[27]
BTV-16e Greece [GRE2008/10]	16e (B)	Yes	No	-	KP820990	[46]
BTV-17w * [RSArrrr/17]	17w (A)	-	-	Yes	AJ585138	[27]
BTV-18w * [RSArrrr/18]	18w (D)	-	-	Yes	AJ585139	[27]
BTV-19w * [RSArrrr/19]	19w (F)	-	-	Yes	AJ585140	[27]
BTV-20e * [RSArrrr/20]	20e (A)	-	-	Yes	AJ585141	[27]
BTV-21e * [RSArrrr/21]	21e (C)	-	-	Yes	AJ585142	[27]
BTV-22w * [RSArrrr/22]	22w (G)	-	-	Yes	AJ585143	[27]
BTV-23e * [RSArrrr/23]	23e (D)	-	-	Yes	AJ585144	[27]
BTV-24 * [RSArrrr/24]	24w (A)	-	-	Yes	AJ585145	[27]
BTV-25 Switzerland (TOV **)	25 (K)	Yes	Yes	-	EU839840	[10]
BTV-26 Kuwait [KUW2010/02]	26 (K ^^)	Yes	Yes	Yes	HM590642	[42]
BTV-27 Corsica (2015) Strain 379	27 (K)	Yes	Yes	-	KM200718	[50]
BTV-27 [COR2014/01]	27 (K)	-	-	Yes	KU760988	[51]

* BTV reference strain. ** Data concerning BTV isolates held in the orbivirus reference collection (ORC) can be obtained at https://www.reoviridae.org/dsRNA_virus_proteins/ReoID/BTV-Nos.htm (accessed on 21 March 2014). ** Toggenberg orbivirus. ^ The nucleotypes of BTV serotypes/isolates are based on phylogenetic analyses of BTV genome segment 2 [24,42]. ^^ BTV-26 was previously assigned to nucleotype ‘L’ [42]. However, based on data presented here showing relationships between the novel serotypes, we have amalgamated nucleotypes K and L and included BTV-25, -26 and -27 in nucleotype K.

### 2.3. Animals 

All animal studies (antiserum production) were performed in the animal facilities at TPI, using ten 14-week-old female New Zealand white rabbits. Throughout the study, daily health checks were performed, and supplemental environmental enrichment was provided (see also Institutional Review Board Statement).

#### Rabbit and Sheep Polyclonal Antisera 

Polyclonal rabbit antisera were raised against ten of the plant-expressed recombinant BTV-VP2 (rVP2) proteins (Table 1). Each inoculum consisted of freshly prepared, purified rVP2 protein, at a concentration of 250 μg/mL, with 500 μL Montanide ISA V50 (Seppic; Colombes, France) adjuvant (*v*/*v*), in a total volume of 1 mL PBS. Inocula were vortexed to mix, until a stable homogenous emulsion was formed, then stored on ice. Each inoculum was administered to a single rabbit. Each animal received subcutaneous vaccinations on days 0, 15 and 32, a total of 1 mL on each occasion, which was split across 4 different injection sites (250 μL each site), giving a cumulative final total by day 32, of 3 mL per animal. At day 46, animals were humanely culled by an overdose of anaesthesia and blood was collected via a cardiac bleed directly into red-top serum blood vacutainers (from BD; Vaud, Switzerland) without anti-coagulant. The blood was allowed to clot for 1 h at room temp, then at 4 °C overnight. Serum was collected and stored at −20 °C.

A panel of BTV reference sheep-antisera, raised against the reference field-strains of BTV serotypes -1 to -24, and -26, was provided by the Non-vesicular Reference Laboratory (NVRL) at TPI. These antisera were derived from sheep previously infected with the referenced bluetongue viruses identified in Table 1.

### 2.4. Serological Assays

#### 2.4.1. Antibodies

Polyclonal rabbit antisera, raised against individual BTV-rVP2 proteins, were purified using the NAb Protein A Plus Spin Kit (ThermoFisher Scientific; Waltham, MA, USA) as per manufacturer’s instructions. Antibody concentration (mg/mL) was determined by spectrophotometer (Eppendorf; Stevenage, UK) at an absorbance of 280 nm. Secondary antibodies were obtained from commercial suppliers, diluted and used as follows: goat anti-rabbit IgG H&L (HRP; Abcam, Cambridge, UK), 1:2000; donkey anti sheep IgG (HRP; Sigma Aldrich, Gillingham, UK), 1:5000. 

#### 2.4.2. Indirect-ELISA

An indirect ELISA (I-ELISA) was developed using purified BTV-rVP2 proteins as target antigens, as previously described [23]. Reagents/buffers were used in 100 μL volumes unless specified otherwise. Briefly, 96-well maxisorp or nickel coated ELISA plates (ThermoFisher Scientific; Waltham, MA, USA) were coated with 2 μg/mL of recombinant protein (VP2) in 0.05 M carbonate-bicarbonate buffer (pH 9.6) (supplied as capsules by Sigma Aldrich (Gillingham, UK), dissolved in PBS) sealed and incubated overnight at 4 °C. Control wells were coated with coating-buffer only, or with 2 μg/mL of purified pEAQ-*HT* (EV) only. Plates were washed 3 times with PBS, 0.05% Tween 20, blocked using PBS, 5% BSA for 1 h at 37 °C, then washed again. Test sera were titrated in duplicate, at dilutions of 1:40 to 1:40,960 (rabbit antisera) and 1:10 to 1:10,240 (sheep antisera), in PBS with 5% skimmed milk powder, with species-specific ‘negative’ serum used as controls. Plates were incubated at room temperature on an orbital shaker for 1 h, washed again and the species-specific HRP labelled secondary antibody was added. Plates were covered and incubated for 1 h at room temperature then washed as previously described. An OPD substrate (SIGMAFAST (Sigma Aldrich; Gillingham, UK)) was added to each test well and incubated in the dark for 15 to 30 min then read immediately at 450 nm using a Multiskan FC microplate photometer (ThermoFisher Scientific; Waltham, MA, USA). The OD value of the negative control at each dilution was deducted from the corresponding OD value of the test serum at the same dilution to eliminate background detection. A cut-off value for positive titres was determined as the mean of the negative control plus one standard deviation. The final antibody titre for the test serum was defined as the inverse of the highest dilution, where the mean value for duplicates was equal to or above the cut-off value. 

#### 2.4.3. Serum Neutralisation Test (SNT)

SNTs were performed as previously described [23] using Vero cells. Plates were scored on days 5–7 for the obvious CPE caused by BTV infection (rounding up and detachment of cells), by visual observation using an inverted light microscope. The final reads (day 7) were used to determine antiserum neutralisation titres, as the inverse of the dilution of serum giving a 50% end-point, as calculated using the Spearman Karber method [45].

### 2.5. Antigenic Cartography 

Multi-dimensional antigenic maps were made using antibody titres generated by I-ELISA to quantify and visualise cross-reactivity between rVP2 proteins from different BTV-strains, using the ACMACS website https://acmacs-web.antigenic-cartography.org/ (accessed on 13 March 2017) as described previously [52]. Briefly, a target distance between each serum and virus was calculated by subtracting the Log_2_ of the titre for that virus, from the Log_2_ of the maximum titre for that serum against any of the other rVP2 proteins. An rVP2 protein that reacts at a high titre with an individual serum, therefore has a smaller target distance to that serum and they are placed closer together on a visual map. Conversely, a low antibody titre detected in a reaction with a given rVP2 protein will give a larger target distance. The target distances, which quantify the antigenic relationships between BTV serotypes/topotypes, are denoted as ‘Antigenic Units’ (AU). One AU is equal to a two-fold change in titre of antiserum regardless of the magnitude of the titre.

Multidimensional scaling was used to minimise the differences (sum-squared error) between the ‘target distance’ and ‘map distance’ (how well the map represents the target distances generated). The position of each virus and antiserum is therefore determined by the relationships and position of each antiserum relative to all other viruses. To minimise the sum-squared error, and obtain a map of best fit, multiple random restart optimisations (500 times) were carried out, generating maps in 2 to 5 dimensions. The correlation between target distance and map distance was used to assess the fit of the maps to the data. There was minimal improvement in overall error and fit by increasing dimensions above 3-D, (Appendix A) and maps are therefore presented in 3D only. 

Antigenic distances (units) were calculated for the expressed BTV-rVP2 proteins, using (bAb) data from I-ELISA, with either rabbit anti-rVP2 sera, or sheep anti-BTV reference antisera (Appendix A). However, due to their higher serotype-specificity in SNT, an insufficient number of cross-serotype nAb reactions were detected using either the rabbit or sheep antiserum panels, to support reliable antigenic cartography.

## 3. Results

### 3.1. Antigenic Cross-Reactivity of Rabbit Antisera against rVP2 Proteins by I-ELISA

Rabbit antisera, raised against the plant expressed rVP2 proteins from ten BTV strains (Table 1), were tested by I-ELISA using rVP2 proteins from seventeen BTV strains as target antigens. Widespread and often high titre cross-reactions were observed (Table 2), to the extent that the anti-rVP2 sera for BTV-6w, BTV-8w, BTV-11w and BTV-27, reacted at different titres with all of the BTV rVP2 proteins tested. The remaining rabbit antisera also recognised most of the rVP2 proteins, apart from anti-BTV1e-rVP2, which only reacted with rVP2 of BTV-1e and -1w and -26. 

Seven of the rabbit antisera (against rVP2 of BTV-1w, -1e, -4w, -8w, -11w, -14w and -27) showed the highest antibody titres in I-ELISA with their homologous rVP2 proteins (titres of 640 to 40,960). High titre cross-reactions were also observed between the rabbit anti-rVP2 sera and rVP2 proteins, derived from eastern and western topotypes of the same serotypes, BTV-1e and BTV-1w (titres ≥ 20,480), and to a lesser extent between BTV-4w and BTV-4e (titre ≥ 5120) (Table 2). The remaining three rabbit antisera (against rVP2 of BTV-6w, -25 and -26: highlighted in blue in Table 2) showed highest titres in one or more of the heterologous reactions. The lowest homologous reaction was between rVP2 and anti-rVP2 of BTV-25 at a titre of 640.

In some but not all cases, high titre cross-reactions were also observed in I-ELISA, between rabbit anti-rVP2 sera and rVP2 proteins derived from heterologous serotypes but from within the same nucleotype (Table 1, Figure 1). For example, the anti-BTV11w-rVP2 serum reacted at the same high titre (40,960) with the rVP2 proteins of BTV-11w and BTV-4w, both belonging to nucleotype A (Table 2). In the reverse reactions, the anti-BTV4w-rVP2 serum also cross-reacted at a high titre (20,480) with BTV11w-rVP2, although this was at a lower titre than in the homologous reaction (40,960). The rabbit anti-rVP2 sera and rVP2 proteins derived from BTV-4w and BTV-11w sera also cross-reacted, but at lower titres, with most of the other rVP2 proteins or anti-rVP2 sera from heterologous BTV serotypes and topotypes. 

However, some high titre cross-reactions were also observed between serotypes belonging to different nucleotypes. For example, the anti-BTV4w-rVP2 serum (nucleotype A) reacted at a high titre (40,960) with rVP2 of BTV-1w (nucleotype H). In the reverse reaction the anti-BTV1w-rVP2 serum, which had an homologous titre of 40,960, showed a much lower titre (1280) with rVP2 of VTV4w. 

Although the rabbit anti-BTV1e-rVP2 serum was the least cross-reactive by ELISA, it did react with rVP2 of BTV-1w (titre of 40,960), and at a low titre with rVP2 of BTV-26 (titre of 40). In contrast, the rVP2 protein of BTV-1e was recognised by all of the heterologous rVP2 antisera, therefore showing multiple, one-way cross-reactions. 

The serum against VP2 of the ‘novel’ serotype BTV-27, appeared to be highly cross-reactive (showing the same titre of 10,240) with the rVP2 proteins of BTV-27, -4e, -9w, -10w, -11w, -14w, and -26 (representing four different nucleotypes; Figure 1). Although the anti-BTV8w-rVP2 serum (homologous titre of 40,960) recognised all seventeen expressed rVP2 proteins, the BTV8w-rVP2 protein itself (homologous reaction titre of 40,960) only cross-reacted with the anti-rVP2 sera of BTV-6w (at a titre of 640), BTV-11w and BTV-27 (each at a titre of 160), none of which belong to the same nucleotype, D.

### 3.2. Cross-Reactivity of Rabbit Anti-rVP2 in Serum Neutralisation Tests (SNT) 

SNTs were performed using reference strains (*n* = 26) of BTV-1 to -24, -26 and -27, to assess the titres and serotype-specificity of neutralising antibodies (nAbs) present in the ten anti-rVP2 rabbit antisera (Table 3). In each case, apart from BTV-27, the highest titre nAbs were detected against the homologous BTV serotype, although titres were generally much lower than by I-ELISA (ranging from 1:15 to 1:690). 

Fewer cross-serotype nAb reactions were observed than in the I-ELISA, and most were between viruses belonging to the same nucleotype. For example, the anti-BTV4-rVP2 sera showed low levels of cross neutralisation with BTV-11w, -17w, -20e and -24w (Table 3), while the anti-BTV11-rVP2 sera contained nAbs that also reacted with BTV-4w, -11w -17w, -20e, and -24w, all of which belong to nucleotype A.

Evidence of intra-nucleotype cross-serotype neutralisation was also seen between BTV-14w and BTV-21e, and between BTV-8w and BTV-23e, in nucleotypes C and D, respectively. Although a strain of BTV-25 that would replicate in cell culture was not available for these studies, the antisera raised against rVP2 of BTV-25, -26 and -27, all cross-neutralised BTV-26 and -27 (Table 3). Based on these results and the similarities detected by phylogenetic analyses of VP2 (Figure 1), we have re-grouped these three ‘novel’ strains/serotypes within nucleotype K (previously grouped in nucleotypes K and L) [32,53]. 

A few inter-nucleotype nAb reactions were also detected, using the rabbit anti-rVP2 sera: with evidence of anti-BTV14w-rVP2 s (nucleotype C) neutralising BTV-4w, -20e and -23e (nucleotypes A and D); anti-BTV1w-rVP2 neutralising BTV-14w (nucleotypes H and C, respectively); and anti-BTV27-rVP2 neutralising BTV-17w (nucleotypes K and A, respectively). BTV-26 was the most cross-reactive virus used in SNT, being neutralised to some extent by seven of the ten rabbit antisera, across five nucleotypes (A, C, D, H, K), although anti-BTV26-rVP2 only neutralised BTV-26 and BTV-17w (Table 3). 

Further evidence for the serotype specificity of the anti-BTV-rVP2 sera in SNT, was provided by the neutralisation of reference strains of BTV-1e -1w, -6w and -8w, only by sera against their homologous strain/serotypes, and by the cross-reaction of eastern and western topotypes of the BTV-1 (Table 3). Rabbit antisera were not generated against rVP2 proteins of BTV-2w, -3w, -5w, -7w, -9w, -12w, -13w, -15w, -16e, -18w, -19w and -22w, however no cross-reactive nAbs were detected against these serotypes in any of the rabbit sera that were generated against other serotypes, again suggesting serotype specificity (Table 3). 

### 3.3. Cross-Reactivity of BTV-rVP2 Proteins with Sheep Anti-BTV Reference Antisera in I-ELISA 

Sheep reference-antisera against BTV serotypes -1 to -24 (excluding BTV-7) and BTV-26 (*n* = 24), were tested in I-ELISA using rVP2 proteins as target antigens (Table 4). Overall, these sheep antisera showed fewer cross-reactions, and generally at lower titres (maximum titre of 10,240) than the rabbit anti-rVP2 sera (maximum titres 1:40,960). However, nine of the sheep antisera (anti-BTV-2w, -5w, -8w, -10w, -11w, -16e, -17w, -19w and -20e) still recognised all or most of the 17 rVP2 proteins from different serotypes (Table 4). 

The most cross-reactive protein was rVP2-BTV-1e, which was recognised in I-ELISA by 17 of the 24 sheep antisera; unlike rVP2 of BTV-1w, which was recognised by only eight sheep antisera (against BTV-1w, -2w, -5w, -6w, -10w, -16e, -17w and -20e). 

All of the sheep reference-antisera recognised the rVP2 protein from their homologous BTV serotype and topotype (where available), showing the highest titres in six out of eleven of these reactions. However, four of the reference antisera (anti-BTV-6w, -10w -11w and -27) showed higher titres in reactions with rVP2 of heterologous serotypes. This includes the BTV-6 and BTV-26 reference sera, which showed only very low positive titres (1:20 and 1:10 respectively) with of rVP2 proteins from strains of their homologous serotype/topotype. 

The ovine anti-BTV-1w serum recognised rVP2 proteins of both BTV-1w and (at a lower titre) BTV-1e, and the sheep antisera against BTV-16e reacted with rVP2 of both eastern and western topotypes of BTV-16, although again the titre was higher with the homologous protein of the eastern strain. However, anti-BTV-9w showed a higher titre with VP2 of BTV-9e than with rVP2 of BTV-9w, and rVP2 of BTV-4e was not recognised by the anti-BTV-4w serum. 

The least cross-reactive sheep sera were against BTV-4w, -14w -15w, which only recognised homologous-serotype rVP2 proteins and consequently showed one-way reactions with the more cross-reactive rVP2 proteins (mentioned above). The rVP2 protein of BTV-2w showed an intermediate level of cross-reactivity by ELISA, being recognised by seven of the 24 sheep reference-antisera (against BTV-2w, BTV-5w, BTV-10w, BTV-16e, BTV-17w, BTV-19w and BTV-20e). However, the anti-BTV-2w sheep serum reacted with all 17 of the available rVP2 BTV proteins (Table 5), providing additional evidence for one-way cross-reactions in ELISA with rVP2 of ten BTV strains.

### 3.4. Cross-Reaction of Sheep BTV Reference-Antisera in SNT with BTV Reference Strains

In contrast to the results from I-ELISA, most reactions detected in SNT with the sheep reference-antisera, were serotype-specific and consistently at higher titres (60 up to 10,000) than observed in SNT with the rabbit anti-rVP2 sera (titres of 15 up to 600: Table 3). This reversed the trend seen in the I-ELISA, which showed generally lower bAb titres in the sheep reference-antisera, as compared to the rabbit anti-rVP2 sera (Table 2 and Table 3). 

Only the sheep anti-BTV-14 serum showed a slightly lower SNT titre with its homologous virus (1:560) than the rabbit anti-rVP2 sera against the same serotype (1:600). Unsurprisingly (since the ‘gold-standard’ for BTV-serotype determination is cross-neutralisation by antisera against reference isolates of the same serotype [24,27,35]), the sheep anti-BTV-1w reference-serum cross-neutralised both topotypes of BTV-1 (east and west), although the heterologous reaction to BTV-1e was at a lower titre (1:40) than with the homologous topotype (1:240). 

Although some cross-reactions were observed in SNT with the sheep sera, all of them were at lower-titres than those recorded in reactions with the homologous strain of the same serotype, unlike those with the rabbit anti-rVP2 sera (compare Table 3 and Table 5). The most cross-reactive BTV strain was BTV-26, assigned to nucleotype K, although the nAb titres detected were low (against 7 other serotypes), even with its homologous antiserum (titre of 60). The most cross-reactive sheep reference-serum was against BTV-3w (homologous titre 560), which neutralised four heterologous serotypes in SNT, although at lower titres ≤ 100. The BTV-4w antiserum, which neutralised its homologous strain (at a titre of 320) and BTV-26 (titre of 1:10), also showed low level nAbs against BTV-17w (titre of 32) and BTV-20e (titre of 32), both of which belong to the same nucleotype as BTV-4w (nucleotype A). However, the BTV-5 antiserum, which neutralised its homologous virus strain and BTV-9w (at titres of 3160 and 10), respectively, both of which belong to nucleotype E, also neutralised BTV-10w (titre of 320) and BTV-26e (titre of 10), which belong to nucleotypes A and K, respectively. 

The least cross-reactive sheep reference-antisera against BTV-10w, BTV-11w, BTV-14w and BTV-16e, only neutralised the homologous virus strains. 

### 3.5. Mapping Antigenic Relationships Using Antigenic Cartography

Antigenic cartography was used to compare and visualise the serological relationships between different BTV strains/serotypes, that were detected in I-ELISA, as multiple cross-reactions by VP2 specific bAbs present in either rabbit anti-rVP2 sera or sheep anti-BTV reference sera, reacting with the expressed rVP2 proteins. The titres of individual antisera (Table 2 and Table 4) were converted into antigenic units (AUs) (shown as Appendix A) and used to position the rVP2 proteins from different virus strains relative to each other in 3D maps (Figure 2). The much higher serotype-specificity of nAbs in SNT with either the rabbit or sheep antiserum panels (Table 3 and Table 5) resulted in an insufficient number of cross-serotype reactions to support reliable antigenic cartography of the different BTV reference strains. 

Overall, both sets of the bAb results (for rabbit or sheep antisera) mapped the rVP2 proteins of different BTV strains/serotypes as a single large cluster (Figure 2), reflecting multiple cross-reactions and antigenic relationships between them. Both maps showed considerable antigenic distances between different topotypes of the same serotype: BTV-1e and -1w (shown in green in Figure 2), or BTV-4e and -4w (shown in purple), although somewhat closer relationships were detected between BTV-9e and 9w (shown in red), or BTV-16e and -16w (shown in orange). 

Cross-reactive nAbs were also detected between BTV-1e and -1w, using both sheep and rabbit antisera in SNT (Table 3 and Table 5), reflecting a relatively close phylogenetic relationship between their VP2 aa sequences (Figure 1). The cross-topotype, intra-serotype reactions of nAbs were not tested for BTV-4, -9 or -16.

The rVP2 proteins derived from BTV strains/serotypes belonging to nucleotype A (BTV-4e and BTV-4w, BTV-10w, BTV-11w) were grouped in both of the I-ELISA maps (shown in purple in Figure 2) reflecting serological (bAb) relationships between them. The nAb results (Table 3 and Table 5) also showed multiple nAb cross-reactions between viruses belonging to nucleotype A (BTV-4w, BTV-11w, BTV-17w, BTV-20e, BTV-24w), again reflecting relatively close phylogenetic relationships detected between their VP2 aa sequences (Figure 1). 

The rVP2 proteins from BTV-26 and BTV-27 (nucleotype K) were positioned relatively closely to those of BTV-4e and -10w (nucleotype A) in Figure 2A (in reactions with rabbit anti-rVP2 sera) and rVP2 of BTV-11w (nucleotype A) was positioned very closely to that of BTV-27 in Figure 2B, possibly suggesting an antigenic (bAb) relationships between these two nucleotypes, which is also indicated by phylogenetic comparisons of VP2 aa sequences (Figure 1). 

The rVP2 proteins of BTV-2w and -8w (nucleotypes D and I, respectively) were placed at the greatest antigenic distances from each other (AU = 10) in the ELISA map based on reactions with the rabbit anti-rVP2 sera (Figure 2A), but were more closely positioned based on reactions with the sheep reference sera (Figure 2B), while the rVP2 proteins of BTV-2w and BTV-1e were placed at greatest distance from each other, suggesting diversity in the bAb responses of different individual animals, or animal-species. 

The level of variations observed in the bAb results obtained by I-ELISA with the two different antisera panels (from sheep and rabbits—Table 2 and Table 4) was greater than the differences observed between the two sets of nAB results from SNT (Table 3 and Table 5). All of the virus strains were positioned differently in the 3D cartography maps of bAb responses (Figure 2A,B) with greater overall antigenic distances calculated for reactions with the sheep antisera, than the rabbit antisera. However, this was likely influenced by the availability of sheep reference antisera for a greater number of different BTV serotypes/strains. The BTV strains that showed the most consistent antigenic relationships by I-ELISA, using both serum panels were: BTV-1w, -9w, -10w, -11w and -16e (difference in AU < 1, between the rabbit and sheep maps).

## 4. Discussion

### 4.1. Serological Reactions between BTV Strains

VP2-bAbs (which include both nAbs and non-neutralising antibodies) were detected at high titres in most of the rabbit and sheep antisera in I-ELISA with homologous rVP2 proteins (titres of 640 to 40,960 or 10 to 10,240, respectively). These homologous assay results provide a value for comparison of the cross-reactions detected against rVP2 proteins of the heterologous viruses. The VP2-bAbs, particularly those in the rabbit anti-rVP2 sera, showed multiple high-level cross-reactions with VP2 proteins of the other BTV serotypes tested, indicating the presence of widely shared epitopes on VP2, many of which are likely to be non-neutralising. The higher bAb titres detected may also reflect the use of exactly the same rVP2 proteins as ELISA antigens, that were used to produce the rabbit antisera, while the sheep sera were raised against ‘reference’ isolates from different geographic origins and earlier isolation dates.

The nAbs detected in the rabbit and sheep antisera (with titres of 10 to 690, or 60 to 10,240, respectively, against the homologous virus serotype and topotype), were more serotype-specific, particularly in the sheep reference antisera that have been used for many years (at TPI) to identify the serotype of novel BTV field isolates. The nAb responses detected in the rabbit sera were generally lower than in the sheep sera, although still primarily against the homologous serotype. This difference may again reflect the use of more recent BTV strains from Europe and elsewhere, as a source of the Seg-2 sequences used to generate the rVP2 proteins, while the sheep reference sera were generated by infections with the same reference strains used in the SNT. The solubility of the plant-expressed proteins and their ability to raise nAbs, which are thought to react primarily with conformation epitopes [22,54,55], as well as previous demonstrations that they can assemble as part of virus-like particles [44], collectively indicate that a significant proportion of the BTV-rVP2 proteins have a native conformation when expressed in plants.

Earlier studies of neutralising antibodies and clinical observations from vaccination/challenge studies in sheep or small animals, identified low-level, one-way or two-way cross-reactions between a limited range of BTV serotypes [56,57,58,59,60]. Some of these relationships were unexpected, based on the magnitude of the amino acid sequence variations in VP2 between individual strains [61]. However, the results obtained may have been biased by use of only a limited range of monoclonal antibodies that recognised conserved sites shared between the different BTV strains tested [55,62].

Cross-serotype nAb reactions were also detected here using a wider range of polyclonal rabbit and sheep antisera, mostly between BTV serotypes within the same Seg-2 nucleotype [24,27], many of which mimic relationships observed in earlier studies of cross-protection between BTV isolates, in sheep vaccination and challenge experiments [37] (Figure 3). One-way reactions were also detected here between viruses or proteins assigned to different nucleotypes, indicating the presence of more widely ‘conserved’ neutralising epitopes.

Variations observed in the overall specificities and titres of bAbs and nAbs, between the rabbit and sheep antisera against individual BTV strains/rVP2 proteins, may in part reflect differences in the immune mechanisms and responses by different mammalian species [57]. However, the rabbits were vaccinated with the expressed and non-infectious VP2 proteins. More cross-reactive epitopes may have been exposed in these individual proteins that could be shielded in whole virus (e.g., by interactions with VP5 and VP7). In contrast, the sheep experienced a full BTV infection, presenting VP2 along with the other viral proteins, as part of an intact and replicating virus. The nAb responses to infection could be influenced by VP2 interactions with the other BTV structural proteins (particularly VP5 and VP7) in the virus capsid. The smaller BTV outer-capsid protein, has been shown to influence the overall specificity of nAbs and consequently BTV serotype, although there is little evidence for direct binding of nAbs to VP5 [4,22,34,36,54,63,64,65]. BTV infection of cellular components of the sheep immune system, and the resulting leucopoenia, may also help to refine the nAb response to infection, enhancing serotype specificity [66].

Large differences were observed in the titres of bAbs or nAbs detected in rabbit and sheep antisera against the homologous viruses or BTV-rVP2 proteins. This suggests that there could also be significant differences in both the magnitude and possibly the cross-reactivity between serotypes, in responses by individual sheep or rabbits to the same antigens. These variations could be further explored in a larger study involving infection of multiple animals with each virus, and vaccinations of multiple animals with the same rVP2 protein.

### 4.2. Multidimensional Mapping of Antigenic Relationships

Using antigenic-cartography to compare antibody cross-reactions and titres provides an advanced visual model in some ways similar to the serological-relationship map developed by Erasmus (1990) [35] (Figure 3). Three-dimensional antigenic maps can help to identify relationships that could be missed by simple analyses of antibody titres [67,68]. The 3D maps shown here indicate a broad range of cross-reactivities within and between BTV serotypes and nucleotypes, clustering the VP2 proteins of all BTV serotypes in one large group, reflecting their common evolutionary origin as well as similar structural and functional roles. The nAB reactions in SNT were more highly serotype-specific than the bAb in I-ELISA, with less cross-reactivity (particularly using the post-infection sheep antisera) implying that the interactions within the sub-set of neutralising antibodies are less complex.

The relative positions of individual strains from different serotypes and nucleotypes in the 3-D maps, could help to inform development of serotyping assays. The relationships detected in SNT could help to predict or select different VP2 proteins or BTV strains for development of polyvalent vaccines [69]. Previous phylogenetic analyses of nucleotide or amino acid sequences of Seg-2/VP2 have provided valuable information concerning the evolution and relationships of BTV strains belonging to different topotypes, serotypes and nucleotypes [24]. Ultimately, variations in the antigenicity of different BTV strains are the result of accumulated mutations in Seg-2 and reassortment of genome segments between strains (genetic drift and shift), although not all mutations will alter the antigenic properties of the individual proteins or virus. Consequently, there is only a partial correlation between sequence variation in Seg-2 and the specificity of the bAbs and nAbs generated. Indeed, the antigenic comparisons and analyses described here show that differences in the amino acid sequence of VP2 between serotypes do not always correlate closely with changes in the overall antigenicity of VP2, as illustrated by the 3D maps.

The differences in sequences and antigenicity between different VP2 topotypes within the same BTV serotype can affect the nAb titres detected. This suggests that the accumulation of mutations and differences resulting in antigenic drift between VP2 topotypes within the same serotype but from different geographic origins may be an important step in the emergence of new and distinct BTV serotypes.

Predictions of antigenic sites based on hydrophobicity analyses indicate that neutralising epitopes may be located towards the amino external-tip domain predicted for VP2 of BTV-1, which is considered likely to be exposed towards the outer surface of the virus particle [70]. This region of VP2, which was not resolved in structural predictions, is also thought to contain neutralising epitopes [36,62]. Previous studies have indicated that VP2 of the insect transmitted orbiviruses have evolved by concatemerisation (duplication) of a smaller cell-attachment protein, as seen in the tick-borne orbiviruses [71]. Phylogenetic analyses of the two separate halves of VP2, generated trees with some obvious differences (Appendix A). The potential for reaction of non-neutralising antibodies with the carboxy terminal half of VP2 might help to explain the wider cross-reactions detected by ELISA, that do not mimic the cross-reactions detected by SNT.

A reaction was detected in both directions using the rabbit anti-rVP2 sera between BTV-26 and -27, weaker reactions were also detected between BTV-25 and BTV-27, supporting the grouping of all three ‘novel’ serotypes in nucleotype K (Figure 1 and Figure 3). Unfortunately, an isolate of BTV-25 that will grow in cell-culture was not available during this study. It was therefore only possible to explore one-way reactions and the ability of anti-BTV-25 sera to neutralise different BTV serotypes (but not vice versa). An update to the serological map originally developed by Erasmus (1990), is shown in Figure 3, illustrating additional two-way relationships that were detected using the expressed rVP2 proteins and rabbit antisera. A relationship was detected between nucleotypes A and K in SNT using the rabbit antisera (Figure 2B), indicated by one-way cross reactions with the novel serotypes (25, 26 and 27) in Figure 3, which may have some evolutionary significance [35]. Several other one-way cross-reactions were also observed in SNT (See Table 3 and Table 5) most of which also mimic relationships detected between nucleotypes by Erasmus [35] but for clarity these are not shown in Figure 3.

Broadly cross-reactive nAbs have previously been detected after sequential vaccination/infection of sheep with modified live vaccines from two different BTV serotypes (BTV-3 and -4) and these were significantly enhanced (and at an accelerated rate) following challenge with a third heterotypic serotype (BTV-6) [72]. The production of cross-reactive nAbs, even at low titres, could potentially prime an enhanced secondary immune response to infection by other heterologous strains (anamnestic immune response), resulting in faster nAb proliferation and greater protection. It may therefore be possible to elicit a broader cross-serotype neutralizing and protective response by combining rVP2 subunits from multiple BTV serotypes as part of simultaneous or sequential vaccinations. A multivalent and sequential vaccination strategy with multiple ‘live’ BTV serotypes was previously used in South Africa [73,74]. The rVP2 proteins described here represent potential subunit vaccine components, removing any risk of incomplete attenuation or subsequent reassortment with field strains associated with modified live-virus vaccines.

The rVP2 proteins of BTV-4 and BTV-8 were previously shown to raise serotype-specific protective immune responses in IFNAR (−/−) mice, thought to reflect generation of serotype-specific nAbs [23]. However, a significant cross-reactive but non-protective bAb response was also detected in the vaccinated mice. The nAb responses detected here against a wider range of heterologous strains/serotypes in SNT may also be cross-protective. Although the cross-reactive bAb responses detected in I-ELISA may be largely non-protective, they could play a role in induction of antibody-dependent cellular cytotoxicity and/or opsonisation [75]. However, the existence of non-neutralising antibodies also raises the possibility of antibody-dependent enhancement (ADE) of infection that could facilitate early dissemination of infection within the mammalian host [76].

The detection of multiple cross-serotype bAb responses suggests that it would be difficult to develop serotype-specific ELISA to identify individual BTV serotypes [58,77]. However, greater serotype-specificity might be possible by expressing a sub-set of serotype-specific VP2 epitopes, rather than the whole VP2 proteins, or by using serotype-specific monoclonal antibodies in a competition ELISA format [78,79].

## 5. Conclusions

BTV-rVP2 proteins can be rapidly and efficiently synthesised in plants from multiple BTV strains. This has made it possible to evaluate immune responses to VP2 in the absence of the other viral proteins, to explore the complex antigenic relationships that exist between multiple different BTV topotypes, serotypes and nucleotypes. The plant-expressed rVP2 proteins raised nAbs in rabbits, although these tended to be at lower titres than in the sheep reference antisera. However, the VP2-bAb response in rabbits was at higher titres and was more cross-reactive than in the sheep reference. The results presented here could be usefully extended and enhanced using a more complete panel of proteins and antisera, representing additional BTV serotypes and their topotypes, as well as multiple animals for each virus/VP2 antigen.

A larger study might also identify sufficient cross-serotype reactions that would support meaningful antigenic cartography of the nAb responses detected in SNT. However, this technique does provide a quick and easy method to visualise and interpret antigenic relationships between VP2 proteins, which were detected as complex bAb cross-reactions in I-ELISA between different BTV strains. The results presented here provide a useful insight into strain cross-reactivity, which may be relevant to vaccine design and serological assay development [45].

## Figures and Tables

**Figure 1 viruses-13-01455-f001:**
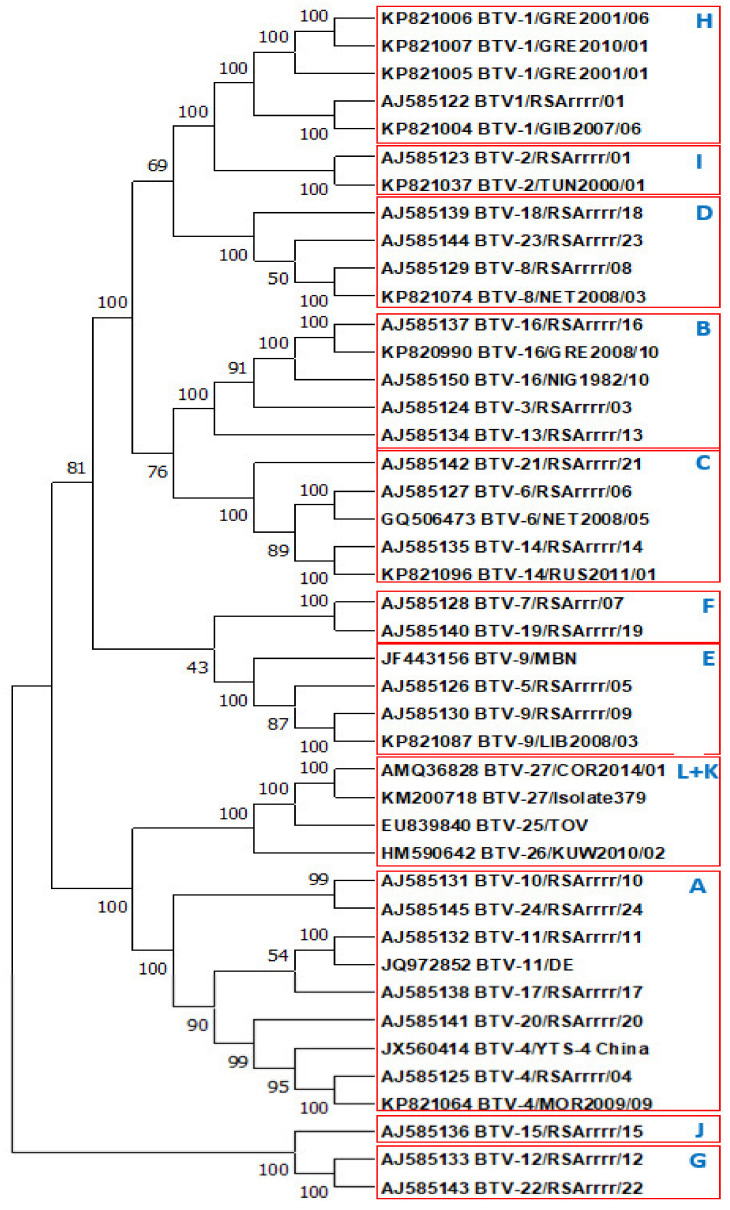
Phylogenetic tree illustrating relationships between VP2 aa sequences of different BTV serotypes. Figure 1: A neighbour-joining phylogenetic tree constructed with aa sequences of VP2(OC1) of BTV-1 to BTV-27 depicting phylogenetic groupings. The tree was generated using the p-distance algorithm (pairwise deletion) implemented in the MEGA X software program. VP2/Seg-2 nucleotypes are indicated as previously reported [41] although with novel serotypes BTV-25, -26 and -27 are included in nucleotype (K).

**Figure 2 viruses-13-01455-f002:**
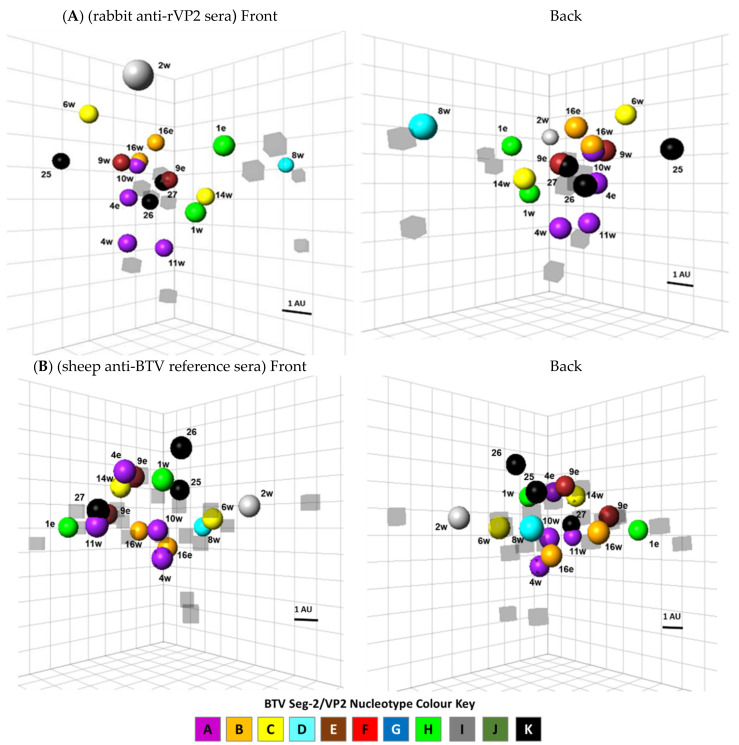
Three-dimensional antigenic maps illustrating cross-relationships between BTV VP2 proteins. Figure 2: Computational multidimensional scaling, with multiple repeat simulations, were used to position the seventeen rVP2 protein antigens derived from different BTV strains and topotypes, in 3-dimensional maps, based on their antigenic relationships as determined by I-ELISA, using either (**A**): the ten rabbit anti-BTV-rVP2 sera (Table 2) or (**B**): the 24 sheep anti-BTV reference sera. The coloured spheres representing rVP2 proteins from different BTV serotypes and topotypes (as indicated) are colour-coded by nucleotype (see colour key). The grey cubes represent either the ten rabbit sera (panel A) or the 24 sheep sera (panel B). The scale bar represents one antigenic unit (AU), equivalent to a two-fold change or difference in antibody titre (Appendix A, for the rabbit and sheep sera respectively). The proteins showing closer antigenic relationships, are therefore positioned closer to each other in the maps. ‘Front’ and ‘Back’ (rotated 180°) 3D views are shown.

**Figure 3 viruses-13-01455-f003:**
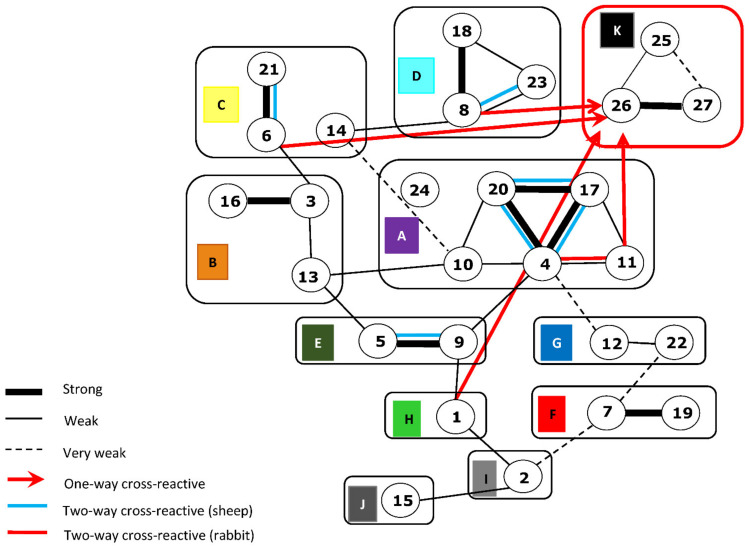
Antigenic relationships between BTV serotypes: The BTV serotype antigenic map, originally developed by Erasmus (1990) [35] to summarise relationships between BTV serotypes, has been modified to include two-way cross-reactions in SNT, detected here with the rabbit anti rVP2 sera (in red) and sheep reference-antisera (in blue). A strong, two-way cross-reaction is shown by a thick black connecting line in the original map, weak reactions by a thin black connecting line and very weak reactions by a dotted black connecting line. The novel BTV serotypes BTV-25, -26 and -27 are included as nucleotype K (red box). The red arrows indicate one-way reactions by rabbit anti-rVP2 sera with BTV-26. Several other one way cross-reactions were also detected in the current study with the rabbit or sheep sera but are not shown in this figure.

**Table 2 viruses-13-01455-t002:** I-ELISA titres for rabbit anti-BTV-rVP2 sera tested against BTV rVP2 proteins.

BTV Strain Providing Sequence Data for rVP2 Expression	Rabbit Anti-BTV-rVP2 Sera									
	1w	1e	4w	6w	8w	11w	14w	25	26	27
BTV-1w [GIB2007/06]	**40,960**	20,480	40,960	2560	640	10,240	1280	1280	1280	2560
BTV-1e [GRE2001/06]	40,960	**40,960**	320	5120	1280	1280	-	640	640	2560
BTV-2w [TUN2000/01]	10,240	-	80	160	40	1280	40	40	40	640
BTV-4w [MOR2009/09]	1280	-	**40,960**	2560	1280	40,960	2560	1280	2560	5120
BTV-4e China YTS-4	1280	-	5120	2560	320	10,240	320	5120	5120	10,240
BTV-6w [ NET2008/05]	640	-	640	**2560**	160	1280	160	320	320	2560
BTV-8w [NET2008/03]	-	-	-	640	**40,960**	160	-	-	-	160
BTV-9w [LIB2008/03]	1280	-	1280	2560	320	2560	320	2560	2560	10,240
BTV-9e India MBN	2560	-	5120	5120	2560	5120	640	2560	5120	5120
BTV-10w [RSArrrr/10]	1280	-	2560	2560	1280	2560	320	2560	2560	10,240
BTV-11w Germany (BTV-11_DE)	1280	-	20,480	1280	160	**40,960**	640	1280	2560	10,240
BTV-14w [RUS2011/01]	10,240	-	-	5120	1280	2560	**40,960**	2560	2560	10,240
BTV-16w [NIG1982/10]	1280	-	1280	5120	640	2560	320	2560	5120	5120
BTV-16e [GRE2008/10]	1280	-	320	5120	640	5120	1280	1280	2560	2560
BTV-25 Switzerland (TOV)	80	-	640	2560	40	1280	-	**640**	640	5120
BTV-26 [KUW2010/02]	2560	40	5120	5120	2560	10,240	1280	2560	**5120**	10,240
BTV-27 Corsica (379)	2560	-	2560	5120	1280	5120	1280	2560	10,240	**10,240**

Antibody titres in reactions with the homologous rVP2 protein are shown in red, bold and underlined. Titres for homologous reactions that were lower than one or more heterologous reaction are shown highlighted in blue. Boxes indicate the reactions of different virus topotypes, with sera from the homologous serotype. The final antibody titre for the test serum was defined as the inverse of the highest dilution, where the mean value for duplicates was equal to or above the cut-off value.

**Table 3 viruses-13-01455-t003:** Neutralising antibody (nAb) titres of rabbit anti-BTV-rVP2 sera in SNT.

BTV Reference Strain [ORC Number]	Rabbit Anti-BTV-rVP2 Sera									
	1w	1e	4w	6w	8w	11w	14w	25w	26e	27w
BTV-1w [RSArrrr/01]	**140**	60	-	-	-	-	-	-	-	-
BTV-1e [GRE2001/09]	120	**60**	-	-	-	-	-	-	-	-
BTV-2w [RSArrrr/02]	-	-	-	-	-	-	-	-	-	-
BTV-3w [RSArrrr/03]	-	-	-	-	-	-	-	-	-	-
BTV-4w [RSArrrr/04]	-	-	**240**	-	-	15	10	-	-	-
BTV-5w [RSArrrr/05]	-	-	-	-	-	-	-	-	-	-
BTV-6w [RSArrrr/06]	-	-	-	**690**	-	-	-	-	-	-
BTV-7w [RSArrrr/07]	-	-	-	-	-	-	-	-	-	-
BTV-8w [RSArrrr/08]	-	-	-	-	**90**	-	-	-	-	-
BTV-9w [RSArrrr/09]	-	-	-	-	-	-	-	-	-	-
BTV-10w [RSArrrr/10]	-	-	-	-	-	-	-	-	-	-
BTV-11w [RSArrrr/11]	-	-	20	-	-	**120**	-	-	-	-
BTV-12w [RSArrrr/12]	-	-	-	-	-	-	-	-	-	-
BTV-13w [RSArrrr/13]	-	-	-	-	-	-	-	-	-	-
BTV-14w [RSArrrr/14]	40	-	-	-	-	-	**600**	-	-	-
BTV-15w [RSArrrr/15]	-	-	-	-	-	-	-	-	-	-
BTV-16e [RSArrrr/16]	-	-	-	-	-	-	-	-	-	-
BTV-17w [RSArrrr/17]	-	-	70	-	-	30	-	-	-	20
BTV-18w [RSArrrr/18]	-	-	-	-	-	-	-	-	-	-
BTV-19w [RSArrrr/19]	-	-	-	-	-	-	-	-	-	-
BTV-20e [RSArrrr/20]	-	-	160	-	-	80	10	-	-	-
BTV-21e [RSArrrr/21]	-	-	-	-	-	-	10	-	-	-
BTV-22w [RSArrrr/22]	-	-	-	-	-	-	-	-	-	-
BTV-23e [RSArrrr/23]	-	-	-	-	30	-	10	-	-	-
BTV-24w [RSArrrr/24]	-	-	15	-	-	10	-	-	-	-
BTV-26e [KUW2010/01]	-	30	-	20	10	20	-	30	**40**	40
BTV-27w [COR2014/01]	-	-	-	-	-	-	-	10	15	**15**

Antiserum neutralisation titre is defined as the inverse of the dilution of serum giving a 50% end-point, using the Spearman Karber method [45]. Titres of nAbs against a strain of the homologous serotype and topotype are shown in red, bold and underlined. The box indicates cross-reactions between different topotypes of BTV-1. The absence of detectable neutralising antibody titres (<1:10) is shown by a ‘dash’. Titres that were lower in the homologous reaction than in one or more heterologous reactions are shaded in blue (BTV-27).

**Table 4 viruses-13-01455-t004:** I-ELISA titres for the sheep reference antisera against BTV-rVP2 antigens.

Sheep Reference Antisera *	BTV rVP2 Antigens																
	1w	1e	2w	4w	4e	6w	8w	9w	9e	10w	11w	14w	16w	16e	25	26	27
BTV-1w	**5120**	1280	-	-	-	40	-	-	-	-	-	-	-	-	-	-	-
BTV-2w	10	160	**10,240**	80	40	10,240	80	20	40	160	20	160	80	40	160	160	20
BTV-3w	-	20	-	10	-	80	-	-	-	-	-	-	-	-	20	-	-
BTV-4w	-	-	-	**10,240**	-	-	-	-	-	-	-	-	-	-	-	-	-
BTV-5w	10	60	20	80	40	320	80	40	80	160	20	80	20	160	320	80	20
BTV-6w	5120	10	-	-	-	**20**	-	-	-	-	-	-	-	-	-	-	-
BTV-8w	-	40	-	80	20	640	**2560**	20	40	80	-	-	160	320	640	80	-
BTV-9w	-	-	-	640	-	-	-	**5120**	10,240	-	-	-	-	-	-	-	-
BTV-10w	320	1280	1280	5120	320	2560	1280	640	1280	**1280**	320	160	320	5120	640	1280	320
BTV-11w	-	40	-	80	20	320	40	10	20	160	**40**	10	-	-	80	10	20
BTV-12w	-	20	-	-	-	40	-	40	80	-	-	-	-	-	80	-	-
BTV-13w	-	10	-	-	10	-	-	-	10	-	-	-	-	-	-	-	-
BTV-14w	-	-	-	-	-	-	-	-	-	-	-	**1280**	-	-	-	-	-
BTV-15w	-	-	-	-	-	-	-	-	-	-	-	160	-	-	-	-	-
BTV-16e	40	80	40	320	40	320	160	40	40	320	40	-	160	**2560**	80	40	40
BTV-17w	20	40	160	320	80	80	20	40	40	320	40	80	160	320	80	20	80
BTV-18w	-	-	-	10	-	-	-	80	-	-	-	-	-	-	-	-	10
BTV-19w	-	20	10	10	10	40	20	320	-	80	20	-	-	-	20	10	20
BTV-20e	10	40	40	1280	20	80	40	20	20	80	-	-	20	40	20	20	40
BTV-21e	-	10	-	10	-	-	-	40	10	-	-	40	-	-	-	-	20
BTV-22w	-	-	-	20		-	-	-	-	-	-	-	-	-	-	-	-
BTV-23e	-	-	-	-	-	-	-	160	-	-	-	-	-	-	-	-	10
BTV-24w	-	10	-	80	-	-	-	160	-	-	-	-	-	-	-	-	10
BTV-26	-	10	-	-	-	-	-	1280	-	40	-	-	-	-	-	**10**	-

Titres for BTV reference antisera, detected in I-ELISA using expressed rVP2 proteins as target antigens. The titres detected against rVP2 proteins from strains of the homologous serotype and topotype, are shown in red, bold and underlined. The boxes indicate homologous serotype reactions for different topotypes within the same serotype. Titres for homologous reactions that are lower than one or more heterologous reaction are shaded in blue. The final antibody titre for an individual test serum in reaction with a specfic rVP2 antigen, is defined as the inverse of the highest dilution, where the mean value for duplicates was equal to or above the cut-off value. * The reference sheep antisera for BTV-7 failed to recognise the homologous reference virus strain, or any other BTV strain tested, and it has therefore been excluded from this study.

**Table 5 viruses-13-01455-t005:** nAb titres of sheep BTV reference-antisera in SNT using reference strains of BTV-1 to -24, -26 and -27.

BTV Strain *	Sheep Anti-BTV Reference Sera **																							
	1w	2w	3w	4w	5w	6w	8w	9w	10w	11w	12w	13w	14w	15w	16e	17w	18w	19w	20e	21e	22w	23e	24w	26
BTV-1w	**10,240**	-	-	-	-	-	-	-	-	-	-	-	-	-	-	-	-	-	-	-	-	-	-	-
BTV-1e	**40**	-	-	-	-	-	-	-	-	-	-	-	-	-	-	-	-	-	-	-	-	-	-	-
BTV-2w	-	**160**	-	-	-	-	-	-	-	-	-	-	-	-	-	-	-	-	-	-	-	10	-	-
BTV-3w	-	-	**560**	-	-	-	-	-	-	-	-	-	-	-	-	-	-	-	-	-	-	-	-	-
BTV-4w	-	-	-	**320**	-	-	-	-	-	-	-	-	-	-	-	5	-	-	32	-	-	-	-	-
BTV-5w	-	-	-	-	**3160**	-	-	10	-	-	-	-	-	-	-	-	-	-	-	-	-	-	-	-
BTV-6w	3.2	-	10	-	-	**1780**	-	-	-	-	-	32	-	-	-	-	-	-	-	32	-	-	-	-
BTV-7w	-	-	-	-	-	-	-	-	-	-	-	-	-	-	-	-	-	-	-	-	-	-	-	-
BTV-8w	-	-	-	-	-	-	**1000**	-	-	-	-	-	-	-	-	-	-	-	-	-	-	10	-	-
BTV-9w	-	-	-	-	10	-	-	**1780**	-	-	-	-	-	-	-	-	-	-	-	-	-	-	-	-
BTV-10w	-	-	-	-	320	-	-	-	**10k**	-	-	-	-	-	-	-	-	-	-	-	-	-	-	-
BTV-11w	-	-	-	-	-	-	-	-	-	**10k**	-	-	-	-	-	-	-	-	-	-	-	-	-	-
BTV-12w	-	-	10	-	-	-	-	-	-	-	**1000**	-	-	-	-	-	-	-	-	-	-	-	-	-
BTV-13w	-	-	-	-	-	-	-	-	-	-	-	**1780**	-	-	-	-	-	-	-	-	-	-	-	-
BTV-14w	-	-	-	-	-	60	-	-	-	-	-	-	**560**	-	-	-	-	-	-	-	-	-	-	-
BTV-15w	-	-	-	-	-	-	20	-	-	-	-	-	-	**560**	-	-	-	-	-	-	-	-	-	-
BTV-16e	-	-	100	-	-	-	-	-	-	-	-	-	-	-	**320**	-	-	-	-	32	-	-	-	-
BTV-17w	-	-	-	32	-	-	-	-	-	-	-	-	-	-	-	**100**	-	-	5	-	-	-	-	-
BTV-18w	-	-	-	-	-	-	-	-	-	-	-	-	-	-	-	-	**1000**	-	-	-	-	-	-	-
BTV-19w	-	-	-	-	-	-	-	-	-	-	-	-	-	-	-	-	-	**10k**	-	-	-	-	-	-
BTV-20e	-	-	-	3.1	-	-	-	-	-	-	-	-	-	-	-	32	-	-	**100**	-	-	-	-	-
BTV-21e	-	-	-	-	-	32	-	-	-	-	-	-	-	-	-	-	-	-	-	**180**	-	-	-	-
BTV-22w	-	-	-	-	-	-	-	-	-	-	-	-	-	-	-	-	-	-	-	-	**1780**	-	-	-
BTV-23e	-	-	-	-	-	-	32	-	-	-	-	-	-	-	-	-	-	-	-	-	-	**1000**	-	-
BTV-24w	-	-	-	-	-	-	-	-	-	-	-	-	-	-	-	-	-	-	-	-	-	-	**1000**	-
BTV-26 *	-	15	10	10	10	-	-	-	-	-	10	10	-	10	-	-	-	-	-	-	-	-	-	**60**
BTV-27 *	-	-	-	-	-	-	-	-	-	-	-	-	-	-	-	-	-	-	-	-	-	-	-	-

Antiserum neutralisation titre was defined as the inverse of the serum dilution giving a 50% end-point, using the Spearman Karber method [45]. * BTV reference virus strains (Table 1) were used in SNT, as well as the isolates: Greece BTV-1e [GRE2001/09]; BTV-26 [KUW2010/02] and BTV-27 [COR2014/01]. ** The post-infection anti-BTV-7w reference serum failed to react with any of the viruses in these assays (including the homologous strain). It has therefore been omitted from these assays. Titres for homologous reactions are shown in red, bold and underlined. The box shows reactions of the ovine anti-BTV-1w serum with eastern and western topotypes of BTV-1.

## Data Availability

Data and research notebooks related to this research are archived at The Pirbright Institute.

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
