# Peer review of "Serological Cross-Reactions between Expressed VP2 Proteins from Different Bluetongue Virus Serotypes"

_viruses, 2021, doi:10.3390/v13081455_

Round 1
Reviewer 1 Report
Dear Authors,
The paper is very relevant and presents an important alternative for vaccine candidate against Bluetongue virus using rVP2 of different BTV serotypes expressed in plants. Results suggest that rVP2 is expressed, at least in part, in the native conformation eliciting neutralizing antibodies. Furthermore, it shows multiple cross-reactions and antigenic relationships between serotypes in I-ELISA and SNT, bringing new information about cross-reactivity between BTV serotypes, topotypes and nucleotypes that can help in the study of the phylogenetic evolution of the virus and may help in development of efficacious strategy for a multi-serotype vacines.
It is an extremely well argued and elegant presented report.
- There are some few suggestions in the text and I bring some questions that I think could be better clarified in discussion.
1- Considering that there is an important individual variation related to an immune response and particularly in antibody response, what could be the consequence to inoculate only one animal per rVP2? If more animals could be inoculated, the results could be affected in an important way?
2- Although all the rabbits received the same amount of rVP2 protein, considering individual immune response and rVP2 antigenicity variation, the Protein A purified antibody titer of α- rVP2 was probably different between the serotypes. What kind of influence it could have on results, considering that antibody titer was the reference for cross-neutralizing or cross-reaction evaluation?
3- In material and methods, when describing SNT it would be interesting to specify what is considering high, medium and low antibody title because it is addressed in many parts of the paper. It is not clear this classification and would be important to standardize.
- Minor observation:
In lines 198-199 and 202-204, the authors explain the definition of I-ELISA and SNT test titre, respectively:
The final antibody titre for the test serum was defined as the mean of the highest dilution at the cut-off value for duplicates. (lines 198- 199)
Plates were scored for CPE on days 5–7, with the final read used to determine antiserum neutralisation titre as the dilution of serum giving a 50% end-point, using the Spearman Karber method. (lines 202-204)
But in the Tables, inverse of the highest dilution are used (for space-saving) or inverse of the titre considered in the definition above. So I suggest addressing it in tables titles and legends.
All the best.

Reviewer 2 Report
General comments:
The manuscript has many instances where words are hyphenated when it does not appear to be necessary.
To make it easier for the reader to identify whether the authors are talking about a viral strain, an antiserum or a recombinant protein, please use a consistent naming convention for each throughout the manuscript.
Abbreviations are introduced for terms and then introduced again in later sections. For example, nAbs and SNT are introduced on line 70 and reintroduced as abbreviations on line 104
Specific comments:
Material and Methods:
Line 125 – was the media supplemented?
Line 135 – how was CPE monitored? crystal violet staining? under the microscope?
Line 138-141 – The Spearman-Karber method is frequently used and a description is unnecessary.
Line 162 – This sentence is confusing. Was each 1 ml of rVP2 protein divided into four 250 ul parts and injected into four different sites on the rabbit? and this was done 3 times on Day 0, 15 and 32?
Line 165 - What kind of vacutainers?
Line 166 - Spell out temperature.
Line 182 – What was the source for the 96-well plates?
Line 183 - Was the coating buffer made in house or purchased?
Line 198 – Please clarify how the final antibody titer was determined. Do you mean above the cut-off value?
Line 203 – How was CPE scored? When was the final read? On day 7 or when CPE reached a certain point?
Line 212 – A virus doesn’t elicit a high antibody titer from a serum. Please clarify this statement.
Line 225-227 – This information is more appropriate after line 215.
Results:
Line 289 - “. . . strong cross-reactions were observed by I-ELISA with viruses. . .” Viruses are not used in the I-ELISA. Do you mean rabbit antisera against the rVP2 from different viruses?
Line 291 – anti-BTV-11w rVP2 missing a dash
Line 294 – BTV11w-VP2. Please check all serum names and rVP2 names for consistency.
Lines 371+ - Please ensure that the two components in the reactions are identified with a consistent naming convention for antisera and expressed protein to prevent confusion with BTV virus strains
Line 438 – Why is this result not surprising?
Line 454 – phrases are redundant
Line 471 – Please describe results of the mapping of ELISA and SNT results separately since the assays quantify different aspects of antibody binding.
Line 473 – Please clarify this statement. The ELISA doesn’t use virus strains. Do you mean the rVP2 from these virus strains?
What is the definition of close antigenic relationship? Does this method provide a means to identify which target distances are significant?
Line 492 – The methods section says that one AU is a two-fold change in titer not a two-fold dilution. Please clarify
Line 713 – The statement that the study demonstrates that the rVP2 protein is in native conformation is somewhat out of place since direct investigation of protein folding and structure were not part of the study.
TABLE 1:
1st column title – doesn’t match order of strain names
Perhaps it would be clearer to make individual columns for information rather than combining it all in one column.
The term post-infected sheep sera is quite cumbersome. If these are reference sera they could be described as being from sheep infected with reference strain BTV viruses and then called anti-BTV sheep serum throughout the rest of the manuscript.
The ORC# is marked with *** and nucleotype is marked with ** but neither are explained in the footnotes.
The strains put into nucleotype K should have a superscript if they were not previously included in nucleotype K.
For consistency, South Africa should be listed as the country for the reference strains.
Are the accession numbers from GenBank or somewhere else?
Is there a reason that BTV-25 is bolded? Also the reference for BTV-1 South Africa is bolded?
Please either use lines that are the same color and thickness throughout the table or eliminate the lines.
TABLE 2:
The ‘S’ is unnecessary since the label says these are serums.
The table’s title and label over the serums should be consistent. “Rabbit rVP2 BTV antisera” vs “Rabbit anti-rVP2 BTV sera”.
The label “BTV strain origin of rVP2” could be shortened to rVP2 and the strain information could be removed since it was given in Table 1.
Some numbers are in blue text. What do they mean?
The direction to the ORC website does not need to be repeated on each table unless it is a requirement for use.
TABLE 3:
It is mentioned in the text which serums did not react with any of the rabbit antisera. These can then be removed from the table.
ORC number is not necessary in every table.
reference – not capitalized
S for serum not needed
If BTV-27 is highlighted in blue, why is BTV-26e/BTV-27w not highlighted?
In these two cases the homologous reactions are = heterologous reactions not less than.
TABLE 4:
The lack of reactivity of BTV-7 antiserum should be noted in the text not in the table.
Please reword title. “. . .post BTV-infection sheep reference antisera. . .” is awkward.
What do the numbers in blue text mean?
For the BTV-16e sera, the homologous titer 2560 is the highest titer but it is highlighted in blue? While for BTV-26 the homologous titer is 10, which is lower than 2 other heterologous titers but it is not highlighted in blue?
Serotype is misspelled.
TABLE 5:
Type out 1000 instead of 1k.
“S” for BTV-26 antisera is not needed.
FIGURE 1:
The method used to produce this figure does not appear in the methods section.
FIGURE 2 and 3:
Please standardize the figure captions.
